# Chyloperitoneum in Peritoneal Dialysis Secondary to Calcium Channel Blocker Use: Case Series and Literature Review

**DOI:** 10.3390/jcm12051930

**Published:** 2023-03-01

**Authors:** Luca Piscitani, Gianpaolo Reboldi, Angelo Venanzi, Francesca Timio, Annamaria D’Ostilio, Vittorio Sirolli, Mario Bonomini

**Affiliations:** 1Nephrology and Dialysis Unit, Department of Medicine, San Salvatore Hospital, Via Vetoio, 67100 L’Aquila, Italy; 2Department of Medicine and Surgery, Centro di Ricerca Clinica e Traslazionale, CERICLET, University of Perugia, 06126 Perugia, Italy; 3Nephrology and Dialysis Unit, Department of Medicine, S. Maria della Misericordia Hospital, 06156 Perugia, Italy; 4Dialysis Unit, Department of Medicine, Maria SS. dello Splendore Hospital, Viale Gramsci, 64021 Giulianova, Italy; 5Nephrology and Dialysis Unit, Department of Medicine, G. d’Annunzio University, Chieti-Pescara, SS. Annunziata Hospital, Via dei Vestini, 66013 Chieti, Italy

**Keywords:** peritoneal dialysis, chyloperitoneum, calcium channel blocker, chylous ascites, effluent, PD complication

## Abstract

Chyloperitoneum (chylous ascites) is a rare complication of peritoneal dialysis (PD). Its causes may be traumatic and nontraumatic, associated with neoplastic disease, autoimmune disease, retroperitoneal fibrosis, or rarely calcium antagonist use. We describe six cases of chyloperitoneum occurring in patients on PD as a sequel to calcium channel blocker use. The dialysis modality was automated PD (two patients) and continuous ambulatory PD (the rest of the patients). The duration of PD ranged from a few days to 8 years. All patients had a cloudy peritoneal dialysate, characterized by a negative leukocyte count and sterile culture tests for common germs and fungi. Except for in one case, the cloudy peritoneal dialysate appeared shortly after the initiation of calcium channel blockers (manidipine, *n* = 2; lercanidipine, *n* = 4), and cleared up within 24–72 h after withdrawal of the drug. In one case in which treatment with manidipine was resumed, peritoneal dialysate clouding reappeared. Though turbidity of PD effluent is due in most cases to infectious peritonitis, there are other differential causes including chyloperitoneum. Although uncommon, chyloperitoneum in these patients may be secondary to the use of calcium channel blockers. Being aware of this association can lead to prompt resolution by suspension of the potentially offending drug, avoiding stressful situations for the patient such as hospitalization and invasive diagnostic procedures.

## 1. Introduction

Approximately 3.8 million people worldwide suffer from end-stage renal disease (ESRD) requiring dialysis therapy. Peritoneal dialysis (PD), although prescribed for only 11% of patients treated with dialysis [1], currently represents an established home-care, cost-effective renal replacement technique. PD is characterized by a more gradual and continuous clearance of fluid and solute, a better preservation of residual renal function, and similar survival to the more commonly used hemodialysis [2]. The depuration of uremic blood in PD is accomplished through exchanges between the peritoneal capillaries and a fluid (PD solution) that is infused into the peritoneal cavity via an implanted intra-abdominal catheter. 

Arterial hypertension is a common comorbidity in subjects with ESRD, including PD patients [3]. Achieving target pressures is of crucial importance to reduce morbidity and mortality, and the use of calcium channel blockers (CCBs) in hypertension treatment of ESRD is very common. It has been estimated that approximately 70% of patients with ESRD are undergoing CCB therapy [4]. 

PD effluent is normally transparent. Patients on PD may exhibit a cloudy peritoneal fluid, which requires a differential diagnosis, the prime suspect being infectious peritonitis [5,6]. A milky-looking peritoneal fluid may also characterize chyloperitoneum (chylous ascites), an uncommon condition secondary to elevated triglyceride concentration. The use of CCBs has been associated with chyloperitoneum. In the present article, starting with a brief report of four more cases of chyloperitoneum occurring in our patients on PD, we review the literature on this complication, and discuss the possible molecular mechanisms involved in its pathophysiology. 

## 2. Presentation of Cases

### 2.1. Case 1

A 70-year-old female patient suffering from ESRD due to nephroangiosclerosis and on automated PD (APD mode) for 8 years had been treated for arterial hypertension during the last 3 months with manidipine 20 mg daily. She contacted our center alarmed at noting chylous peritoneal drainage fluid in the absence of fever and abdominal pain. In the dialysate, the leukocyte count was negative, the culture test for common germs and fungi was sterile, and triglycerides were elevated (218 mg/dL), with normal cholesterol (19 mg/dL). In the peripheral blood, assays of cholesterol (150 mg/dL), triglycerides (109 mg/dL), lipase (49 U/L), and amylase (26 U/L) were in the normal range. An ultrasound of the abdomen excluded secondary causes. When CCB-associated chyloperitoneum was suspected, manidipine was promptly suspended; on the following day, the drained peritoneal fluid was clear and remained so thereafter (Figure 1). It should be noted in this case that the event occurred 3 months after the start of therapy with manidipine.

### 2.2. Case 2

A 56-year-old female patient with ESRD due to autosomal dominant polycystic kidney disease had been on PD for only 15 days and was therefore in the break-in phase. She was on manidipine 20 mg/day. During the training phase, the peritoneal fluid showed a milky appearance in the absence of fever and abdominal pain. In the dialysate, the leukocyte count was negative and the culture test for germs and fungi was sterile. The decision was to discontinue manidipine and re-evaluate the patient after 24 h. At the follow-up visit, the dialysate was clear and remained so on the following days. Three months after the event, the patient noted a marked increase in blood pressure values, and spontaneously resumed manidipine 20 mg daily for three consecutive days, with the recurrence of chyloperitoneum which was resolved upon discontinuation of manidipine.

### 2.3. Case 3

This eighty-year-old man had ESRD due to IgA nephropathy and a history of hypertension. His usual daily medications included furosemide 500 mg, calcitriol 0.25 mcg, tamsulosin 0.4 mg, simvastatin 10 mg, and ramipril 5 mg. He started PD in December 2021 on the following schedule: three daily exchanges with 1.5% glucose and nighttime icodextrin 3.75%. In March 2022, ramipril was withdrawn because of hyperkaliemia and was replaced with lercanidipine 10 mg/day, one single dose in the evening. After 10 days, the patient came to the clinic reporting that his morning effluent had a milky appearance. He was asymptomatic, afebrile, and hemodynamically stable, and both the physical examination and an abdominal ultrasound were negative. The dialysis effluent white cell count (WBC 24 cell/µL) and culture were both negative. The triglyceride level in the effluent was 48 mg/dL. In the absence of any obvious etiology, a diagnosis of chyloperitoneum was suspected. Given its potential association with chyloperitoneum, lercanidipine was discontinued. Within the following 72 h, the peritoneal fluid was completely normal and clear once again.

### 2.4. Case 4

A 76-year-old woman with ESRD due to nephroangiosclerosis (biopsy proven) and a long-standing history of hypertension (30 years) was being treated with lercanidipine 20 mg/day, nebivolol 5 mg/day, and transdermal clonidine 5 mg/week. In September 2022, she underwent peritoneal catheter implantation and initiated PD. Shortly after starting the training phase, the peritoneal fluid showed a milky appearance in the absence of fever and abdominal pain. At presentation, the patient was asymptomatic, hemodynamically stable, and the physical examination of the abdomen was negative. An abdominal ultrasound did not disclose any abnormal findings. The dialysis effluent white cell count was negative (WBC 31 cell/µL) and the fluid culture was sterile. The triglyceride level in the effluent was 32 mg/dL. In the absence of any obvious etiology, lercanidipine was stopped because of its potential association with chyloperitoneum. The change was followed by the normalization of the peritoneal fluid effluent appearance after 24 h. 

### 2.5. Case 5

This white male, 53 years old, had ESRD following IgA nephropathy (diagnosed in 2015 via renal biopsy) and a history of high blood pressure; he was treated with lercanidipine 20 mg/day and doxazosin 4 mg for three times per day. A peritoneal catheter was inserted in December 2022, and early in the break-in phase the nurse noted a milky-like effluent. Clinically, he was asymptomatic and hemodynamically stable, and the abdomen was treatable and not painful. The peritoneal fluid white cell count (WBC 66 cell/ µL) and culture were negative. The effluent triglyceride level was 37 mg/dL, and the total cholesterol level was <20 mg/dL. Diagnostic imaging excluded neoplasms, cirrhosis, and recent trauma. With no evidence of peritonitis, we suspected chyloperitoneum due to CCBs and immediately withheld lercanidipine. Within 48 h, the effluent fluid was clear.

### 2.6. Case 6

This 54-year-old white woman, with ESRD due to lupus nephritis (diagnosed in 1991 via renal biopsy) had been on peritoneal dialysis since January 2021 using APD (three 5000 cc loads at the following concentrations: one 1.36% and two 2.27%, plus a 1500 cc load of icodextrin during the day). Her blood pressure control was non-optimal and lercanidipine (20 mg/day) was added to carvedilol (6.25 mg twice daily), ramipril (5 mg), and doxazosin (4 mg three times per day). Shortly afterwards, a milky peritoneal effluent was seen in the absence of fever and abdominal pain. Clinically, the patient was asymptomatic and the abdomen was not painful. The effluent culture was negative, with normal cell counts, and a complete abdomen ultrasound showed no abnormalities. We therefore suspended lercanidipine and within 72 h the peritoneal fluid was completely clear.

## 3. Discussion

The appearance and the color of peritoneal fluid are of the utmost importance in clinical practice; the recognition of a peritoneal fluid change can lead to early and appropriate intervention [5]. The color of the peritoneal fluid can help in diagnosis, not only concerning technique-related complications but also underlying pathologies. A milky white dialysate suggests the occurrence of chyloperitoneum, defined as the presence of lymph in the abdominal cavity. Chyloperitoneum is a rare clinical condition that can be traumatic or nontraumatic in origin. Possible causes of chyloperitoneum are reported in Table 1.

Chyloperitoneum in PD patients associated with the use of CCBs is rarely discussed in the literature [7]. Turbid dialysate can be caused by infection, neoplasia, chemical inflammation secondary to drugs and icodextrin, or lymphatic vessel rupture, while more rarely it is secondary to the use of CCBs. Chyloperitoneum secondary to the use of CCBs is also reported in the general population [5,8]. As recently reviewed [7], cohort studies (three retrospective and one prospective) showed the occurrence of chyloperitoneum in PD patients treated with lercanidipine with an average prevalence of 25.97%, and with four out of nine CCBs (manidipine *n* = 15; benidipine *n* = 2; nisoldipine and nifedipine *n* = 1) in the largest study including 251 patients. Re-administration of CCBs was associated with a recurrence of chyloperitoneum. Case series and case reports published from 1993 to 2022 are summarized in Table 2 [6,9,10,11,12,13,14,15,16,17,18,19,20,21,22].

CCBs represent a cornerstone in hypertension therapy. They are liposoluble drugs with a greater ability to bind to the cell membrane. There are three different types of CCB (dihydropyridines, phenylalkylamines, and benzothiazepines) that differ in molecular structure, site of action, and effects on cardiovascular functions. A recent review shows that CCB drugs are associated with chyloperitoneum regardless of the subclass, although those most frequently involved are the dihydropyridine CCBs manidipine and lercanidipine [7]. The time of onset is a few days after CCB initiation in most studies, though it may range from as early as 8 h [22] up to 3 months (Case 1 of the present report).

Chyloperitoneum is characterized by the presence of lymph in the abdominal cavity, due to disruption of the lymphatic flow. Lymph is a mixture of lipids/chylomicrons, proteins, and immune cells with about 50% originating in the intestine [23]. It has a milky appearance due to the high content of long-chain triglycerides, which are absorbed into the lymphatic system as chylomicrons following conversion into fatty acids and monoglycerides in the small intestine. The lymphatic system is responsible for maintaining a proper tissue–fluid balance. Through the lymphatic system, around 8 L of fluid and proteins otherwise accumulating in extravascular compartments is returned daily to the venous circulation [24]. The propulsive force that moves and returns lymph centrally is two thirds caused by the contraction of collecting lymphatic vessels [25]. Smooth muscle cells in the wall of collecting lymphatic vessels may contract rhythmically, with the compression of successive chambers comprising the vessels [26]. Each transient lymphatic contraction is triggered by a pacemaker-generated action potential in lymphatic smooth muscle cells involving spontaneous transient depolarizations [27]. Note that effective lymph propulsion requires not only robust contractions of lymphatic muscle cells but also the coordination of contraction waves along the lymphangion, the segment of a collecting lymphatic vessel between two intra-luminal valves [28].

Concerning the molecular mechanisms of lymphatic contraction, lymphatic muscle cell contraction is primarily dependent on the influx of Ca^2+^ through L-type voltage-gated calcium channels [29,30]. L-type calcium channels (also termed Ca_v_1) are integral membrane proteins which play a critical role in a wide spectrum of physiological processes including excitation–contraction coupling in smooth, cardiac, and skeletal muscle [31]. In response to action potential L-type calcium channels, they undergo conformational transitions and modulate the Ca^2+^-inflow across the membrane [31]. Intracellular Ca^2+^ is essential to lymphatic contractions and to spontaneous transient depolarizations of lymphatic smooth muscle [26]. Ca^2+^ binds to calmodulin forming a complex that activates myosin light chain kinase which in turn phosphorylates regulatory light chain kinase 20 [32]. The phosphorylation of myosin light chain 20 promotes muscle contraction by activating the interaction of myosin with actin [32].

A recent comprehensive review examining the direct effect of a drug on the lymphatic vessel contractile function showed CCBs (among 208 identified drugs from 193 studies) to be the most frequently used drugs with inhibitory effects on lymphatic pump function, usually decreasing contraction frequency [33]. Blocking calcium channels inhibited both spontaneous action potential and the associated contractions in different models of lymphatic vessels [26,34,35]. The activity of L-type calcium channels may be antagonized by 1.4-dihydropyridine drugs [36] through binding to a highly conserved receptor site [37] preferentially in the inactivated state to stabilize L-type channels via the blocked inflow of Ca^2+^. Phenylalkylamines such as diltiazem and verapamil are also antagonists of L-type calcium channels [36], though with a binding affinity for different sites [31]. However, L-type calcium channels display a similar pharmacological profile upon treatment with antagonist drugs [31]. 

Though the mechanism of chyloperitoneum via CCBs in PD patients is not clear, a relation may be suggested with lymphatic dysfunction in triglyceride disposal caused by CCB inhibition of lymphatic vessel smooth muscle. CCBs are lipophilic drugs that can pass rapidly into the lymphatic system, acting on the smooth muscle cells of the gastrointestinal tract and blood and lymphatic vessels [38]. The higher lipophilicity of lercanidipine and manidipine than other CCBs [39] might explain why they are more often reported in the genesis of chyloperitoneum in patients on PD. CCB inhibition of the contractility of lymphatic vessels’ smooth cells may hamper lymphatic drainage, by causing vasodilation and an increase in the hydrostatic pressure of the lymph vessels, ultimately leading to the exudation of lymph through the walls of dilated retroperitoneal vessels into the peritoneum [8]. Considering that healthy lymphatic collecting vessels constitutively leak a portion of the transport (fluid, solutes) into the surrounding tissue [40], it is conjecturable that CCB-induced impairment of the lymphatic barrier function will lead to severe lymph leakage into the peritoneum. In lymphatic muscle, contraction amplitude and force production are consistently reduced by sub-maximal concentrations of L-type blockers [41]. Moreover, therapeutic concentrations (nanomolar) of nifedipine have been shown to inhibit spontaneous phasic contractions and electrical activity in isolated human lymphatic cells [30]. However, this finding was not duplicated in clinical studies on human volunteers which showed that concentrations of nifedipine affecting lymphatic contraction activity in vitro did not affect contractile activity in vitro [30], possibly because the pressure-induced increase in contractile activity overcomes the inhibitory effect [33]. Although these results do not per se exclude an inhibitory effect of nifedipine on the lymphatic vessels, they indicate that the dominant in vivo effect is on the blood vasculature. However, evaluating the effects of a drug on lymphatic contractile function is complicated for several methodological reasons, and it also proves difficult to determine direct cause–effect relationships with the systemic administration of drugs in in vivo experiments [33]. Nifedipine affects the arteries and fluid movement in the capillaries, thereby having an indirect role on the lymphatics. Further studies are thus needed to appropriately investigate the discrepancy between the in vitro and in vivo results reported by Telinius et al. [30].

There are other factors possibly involved in developments in the peritoneum seen in PD patients on CCBs. Hsiao et al. [42] reported that nine out of forty CAPD patients developing chyloperitoneum through the addition of lercanidipine tended to have higher peritoneal membrane transport with an increased amount of effluent. The high solute transport rate may lead to more lercanidipine being accumulated in the peritoneal cavity through the diffusion process, which in turn may decrease lymphatic absorption via its contraction on the lymphatic stomata located in the sub-diaphragmatic peritoneum [43], favoring chyloperitoneum [42]. On the other hand, vasoactive drugs such as CCBs may per se influence the peritoneal transport rate via either intraperitoneal or systemic administration, irrespective of the effects on arterial blood pressure. In experimental PD models, intraperitoneal administration of three different CCBs (nicardipine, diltiazem, and verapamil) dose-dependently increased the peritoneal net fluid absorption rate and the permeability to urea and glucose [44]. Intraperitoneal administration of verapamil or nifedipine significantly increased small solute clearance and ultrafiltration in CAPD patients [45,46]. Furthermore, in a randomized study involving nine hypertensive patients on CAPD, the use of oral nifedipine was associated with a significant increase in creatinine and beta-2 microglobulin peritoneal clearance [47]. These results point to action by CCBs on the arteriolar end of peritoneal capillaries, augmenting the peritoneal vascular surface area and permeability, with an ensuing increase in the peritoneal ultrafiltrate with higher levels of triglycerides.

Finally, patient characteristics predisposing one to CCB-induced chyloperitoneum in PD cannot be ruled out. Besides their individual peritoneal membrane transport properties, patients with a lower baseline lymphatic contractile function may be more prone to CCB-caused edema [33]. An ethnic background was suggested following the observation that most PD patients with CCB-related chyloperitoneum are from Asia [42]. Moreover, polymorphisms of the calcium channel gene might affect one’s susceptibility to chylous ascites [39]. The core pore-forming alpha 1 subunit of L-type calcium channels, which plays the main biophysical and pharmacological roles in the channel, is encoded by four different genes according to the channel isoform [31]. The mutation of the genes which participate in aspects of lymphatic vasculature, even though subtle and undetected, may cause pathological effects in humans [48]. Thus, mutations in the encoding gene of the L-type calcium channel may lead to symptoms of lymphatic dysfunction such as chyloperitoneum upon treatment with CCBs, a hypothesis which is at present only speculative and deserves further investigation.

It thus appears that chyloperitoneum in PD may be related to several not mutually exclusive mechanisms. A better definition of the molecular pathways involved may lead to a possible reduction in the clinically observed development of this complication in PD patients treated with CCBs. 

The present study includes six more patients than reported in a previous systematic review [7], further suggesting that clinicians should carefully evaluate the pharmacologic therapy of a PD patient developing chyloperitoneum. In keeping with previous reports, we found no association between the occurrence of chyloperitoneum and age, gender, or years of peritoneal dialysis duration, nor with serum triglyceride concentration. Hence, there are no known predictive factors. Moreover, the triglyceride concentration threshold for the diagnosis of chyloperitoneum is uncertain because the triglyceride concentration in the ascitic fluid can be diluted by the peritoneal dialysate [42]. The cut-off of 200 mg/dL could, therefore, cause misdiagnosis, underestimating the true incidence of chyloperitoneum in patients on PD.

The treatment of chyloperitoneum depends on the cause [23]. Diet modification can aid in managing it, either through total parenteral nutrition or by removing LCTs from the diet [49]. In the neonatal population, the use of a low-fat formula with MCTs is recommended [14]. Treatment with octreotide appears to have an effect: it blocks intestinal somatostatin receptors by decreasing the intestinal absorption of fatty acids, and reduces intestinal blood flow and lymphatic secretion [50]. However, chyloperitoneum, secondary to the use of CCBs, is completely reversible within 24/72 h, so suspending the offending medication is usually followed by complete resolution with no need for other intervention.

## 4. Conclusions

A change in the appearance of PD effluent indicates a complication which may or may not be related to the dialysis technique itself. The diagnosis and management of chyloperitoneum in PD can be challenging. It may require empiric antibiotic therapy, instrumental examinations, and paracentesis, thereby increasing the workload and healthcare costs. Although several other causes of chyloperitoneum should be considered, we feel it is of paramount importance to highlight the possibility that it may be secondary to the use of calcium channel blockers. Use of this medication class in patients on PD is indeed common, and, although the incidence of chyloperitoneum is rare, being aware of this association can lead to prompt resolution simply by suspension of the potentially offending drug. In addition to the clinical evidence gathered so far, further studies are needed to elucidate the mechanistic and causal pathways of CCB-associated chyloperitoneum in patients on PD. 

## Figures and Tables

**Figure 1 jcm-12-01930-f001:**
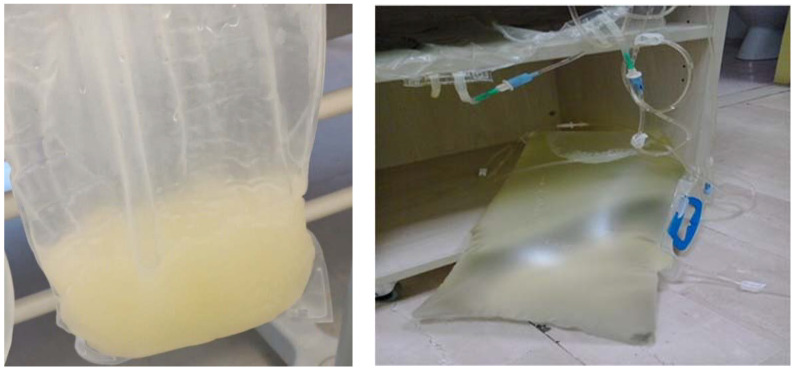
Appearance of peritoneal effluent before (**left panel**) and the day after (**right panel**) the suspension of manidipine. The same variation from cloudy peritoneal fluid with a milky appearance to a normal clear appearance was observed in the other patients of the present report upon withdrawal of the calcium channel blocker drug.

**Table 1 jcm-12-01930-t001:** Causes of chyloperitoneum.

Cause	Mechanism
Trauma Abdominal or thoracic surgery Infections (tuberculosis, filariasis) Autoimmune disease (SLE, sarcoidosis) Retroperitoneal fibrosis Radiotherapy	Obstruction or disruption of the thoracic duct
Cirrhosis Cardiovascular disease (congestive heart failure, constrictive pericarditis, superior vena cava syndrome)	Increased production of lymph with +/− less effective venous drainage
Lymphomas Sarcomas Leukemia Solid organ malignancies	Invasion and disruption of the normal lymph flow
Calcium channel blockers Aliskiren	Drugs
Channel-type lymphatic malformation Yellow nail syndrome Common (cystic) lymphatic malformations Klippel-Trenaunay syndrome	Disruption or dilatation of lymphatic vessels

SLE, systemic lupus erythematosus.

**Table 2 jcm-12-01930-t002:** Main characteristics of published case series and case reports of chyloperitoneum in PD patients.

Study [ref. #]	Patient Characteristics			Drug Name, Daily Dose	Result of Withdrawal	Result of Rechallenge
	Age *	Sex	PD Vintage			
Present report	70	F	8 years	Manidipine, 20 mg	Dialysate clear within 24 h	-
	56	F	15 days	Manidipine, 20 mg	Dialysate clear within 24 h	Cloudy again
	80	M	3 months	Lercanidipine, 10 mg	Dialysate clear within 72 h	-
	76	F	2 days	Lercanidipine, 20 mg	Dialysate clear within 24 h	-
	54	F	1 year	Lercanidipine, 20 mg	Dialysate clear within 48 h	-
	53	M	15 days	Lercanidipine, 20 mg	Dialysate clear within 72 h	-
Figueiredo 2022 [9]	68	M	2 months	Lercanidipine (from nifedipine), NA	Dialysate clear within 24 h	-
Pasquinucci 2021 [6]	61	M	NA	Lercanidipine, 20 mg	Dialysate clear within 24 h	-
Nicotera 2018 [10]	53	F	2 years	Lercanidipine, 20 mg	Dialysate clear (timing NA)	-
Gupta 2016 [11]	65	M	8 days	Amlodipine, 5 mg	Dialysate clear within 24 h	-
Moreiras 2014 [12]	59	F	NA	Lercanidipine, 5 mg	Dialysate clear within 24 h	Cloudy again
Betancourt 2013 [13]	60	M	2 months	Manidipine, NA	Dialysate clear	-
	41	M	4 months	Verapamil, NA	Dialysate clear	-
	70	F	4 days	Manidipine, NA	Dialysate clear	-
	52	M	5 months	Manidipine, NA	Dialysate clear	-
Mallett 2012 [14]	7 mo	M	5 months	Amlodipine, 0.6 mg/kg	Dialysate clear before withdrawal	TG slightly increased in dialysate
Ram 2012 [15]	55	M	1 day	Diltiazem, NA	Dialysate clear after 1 day	Cloudy again
Lopez 2011 [16]	44	F	NA	Manidipine (from nifedipine), NA	Dialysate clear within 24 h	-
Tsao 2009 [17]	41	F	2 weeks	Lercanidipine, 10 mg	Dialysate clear within 24 h	Cloudy again
Roh 1999 [18]	47	M		Manidipine, 40 mg	Dialysate clear after 1 day	-
Tsurusaki 1995 [19]	36	M	32 months	Manidipine, 20 mg	Dialysate clear within 24 h	-
Fujii 1995 [20]	58	M	2 months	Manidipine, 10 mg	Dialysate clear within 24 h	-
Kato 1994 [21]	51	F	4 months	Manidipine, 40 mg from 20 mg	Dialysate clear within 12 h	-
Kugiyama 1993 [22]	44	M	6 months	Manidipine, 20 mg	Dialysate clear within 1 day	Cloudy again

* Years unless otherwise indicated (mo = months). F, female, M, male; h, hours; NA, not available.

## Data Availability

All data are available upon request to the corresponding author.

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
