# Peer review of "Chyloperitoneum in Peritoneal Dialysis Secondary to Calcium Channel Blocker Use: Case Series and Literature Review"

_jcm, 2023, doi:10.3390/jcm12051930_

Round 1
Reviewer 1 Report
This is a short article that reports the occurrence of chyloperitoneum in patients on peritoneal dialysis treated with calcium channel blocker.
Strengths of the manuscript:
– Topic of current interest
– Well-written and well-organized article
– No significant error in the research design.
– Well-conducted discussion
– Correct references
Weaknesses of the manuscript:
No weakness of the manuscript apart from the highly specialized nature of the subject
Itemized list of specific items
1 In the table, lymphangiectasia should be replaced by the term currently recommended in the new nomenclature
2 In the table, lymphangioma au moi should be replaced by the term recommended in the new nomenclature
Reviewer 2 Report
This is an interesting report and the structure is good.
But there are extensive grammatical errors throughout the manuscript which are beyond the scope of a reviewer to correct.
Some examples:
Line 89: Is it assumed or resumed?
Line 90: Please re-write as “ with recurrence of chyloperitoneum which resolved on discontinuation of manidipine”
Line 99: patient came presented to the clinic
Please refer to an English medical writer to revise the manuscript
Reviewer 3 Report
The manuscript entitled "Chyloperitoneum in peritoneal dialysis secondary to calcium-channel blocker use. A rare or unrecognized complication? Case series and literature review" is a case-series study reporting cases of chyloperitoneum due to calcium-channel blocker use observed in six patients affected by end-stage kidney disease treated by peritoneal dialysis. The authors performed a well-done narrative review of reported cases of chyloperitoneum associated with using calcium-channel blockers in those patients, including an exhaustive explanation of the pathogenetic mechanism underlying the onset of the complication.
I suggest only minor revisions:
- Did the authors recognize other risk factors predisposing to the onset of this complication in PD patients (e.g., high transporters, gender, etc.)? Were calcium blockers used as the only risk factor?
- Table 2 needs to be clarified. I suggest highlighting colour-difference among the lines of the table according to the mechanism.
- I suggest changing the title because the Authors cannot answer this study's question (rare or unrecognized complication?).
